# EEG Amplitude Modulation Analysis across Mental Tasks: Towards Improved Active BCIs

**DOI:** 10.3390/s23239352

**Published:** 2023-11-23

**Authors:** Olivier Rosanne, Alcyr Alves de Oliveira, Tiago H. Falk

**Affiliations:** 1Institut National de la Recherche Scientifique, University of Quebec, Montreal, QC H5A 1K6, Canada; olivier.rosanne@inrs.ca; 2Graduate Program in Psychology and Health, Federal University of Health Sciences of Porto Alegre, Porto Alegre 90050-170, Brazil; alcyr@ufcspa.edu.br

**Keywords:** active BCI, mental state, modulation features

## Abstract

Brain–computer interface (BCI) technology has emerged as an influential communication tool with extensive applications across numerous fields, including entertainment, marketing, mental state monitoring, and particularly medical neurorehabilitation. Despite its immense potential, the reliability of BCI systems is challenged by the intricacies of data collection, environmental factors, and noisy interferences, making the interpretation of high-dimensional electroencephalogram (EEG) data a pressing issue. While the current trends in research have leant towards improving classification using deep learning-based models, our study proposes the use of new features based on EEG amplitude modulation (AM) dynamics. Experiments on an active BCI dataset comprised seven mental tasks to show the importance of the proposed features, as well as their complementarity to conventional power spectral features. Through combining the seven mental tasks, 21 binary classification tests were explored. In 17 of these 21 tests, the addition of the proposed features significantly improved classifier performance relative to using power spectral density (PSD) features only. Specifically, the average kappa score for these classifications increased from 0.57 to 0.62 using the combined feature set. An examination of the top-selected features showed the predominance of the AM-based measures, comprising over 77% of the top-ranked features. We conclude this paper with an in-depth analysis of these top-ranked features and discuss their potential for use in neurophysiology.

## 1. Introduction

Active brain–computer interfaces (BCIs) have emerged as powerful communication tools for users with severe and multiple disabilities [1]. In recent years, BCIs have dropped in price and become portable, thus allowing for so-called passive applications to also emerge, e.g., in entertainment, marketing, and mental state monitoring for safety, to name a few [2,3,4]. Medical applications, particularly in neurorehabilitation [5,6], still remain a predominant use of active brain–computer interfaces (BCIs) as they can improve the quality of life for patients suffering from amputations or paralysis due to neuronal damage, such as stroke [7,8]. Typical interventions range from upper limb rehabilitation [9] to gait enhancement [10], communication support [11], and interactive engagement [12] via the modulation of sensorimotor rhythms to aid motor function restoration and drive neuroplasticity [13].

Despite the great potential for both active and passive BCIs, there are still several major challenges that need to be overcome. From the user perspective, psychological state and familiarity with BCI technology influence the efficacy of rehabilitation. Studies have shown that mental state, such as fatigue, frustration, and attention level, can significantly affect BCI performance [14]. Since learning to use a BCI system demands considerable mental effort, user fatigue is a significant psychological factor, underscoring the importance of user motivation in the successful adoption of BCI systems [15]. Furthermore, other factors, such as individual attention span and spatial ability, also contribute to the variable reliability of BCIs in practical scenarios [16,17]. Overall, several major challenges relating to the robustness of BCIs still exist, such as inter- and intra-subject variability [18], as well as varying signal quality (e.g., signals obtained from gel-based versus dry electrodes) and artifacts, which can bury specific task-related brain activity within noise [19,20]. In fact, BCI performance is highly dependent on the settings that the system has been trained for; thus, any out-of-domain test settings can drastically reduce accuracy.

Over the years, several approaches have been proposed to improve the robustness of BCIs. Multimodal systems, or so-called hybrid BCIs [21,22], take advantage of different neurophysiological modalities (e.g., eye and facial movements or hemodynamics via functional near-infrared spectroscopy (fNIRS)) to improve BCI accuracy. Other approaches have included the development of new signal processing and feature extraction tools to help sift out important brain patterns from noise. For example, the last decades have seen developments in features such as fractal dimension and entropy measures [23,24], as well as the use of amplitude envelope-based features [25,26] to complement traditional spectral power [27,28,29], spectral coherence [30,31], and time domain statistics [32] features. More recently, deep learning models have emerged as data-driven methods that can help to advance BCI technologies [33,34,35,36]. While such data-driven methods have shown improved accuracy on specific datasets, they are known to poorly generalize across datasets [35,37] and may introduce new vulnerabilities (e.g., susceptibility to adversarial attacks [38,39]).

In this paper, we propose to improve active BCI robustness by incorporating a new signal representation that allows for the measurement of amplitude modulation (AM) dynamics and cross-frequency coupling using electroencephalography (EEG) signals. While such a representation has been used in the past for Alzheimer’s disease biomarker development [40,41] and passive BCI monitoring [42], it is explored here as a new feature for active BCIs. The proposed method has several advantages over conventional power spectral density (PSD) features [43,44,45], which motivated this exploration. First, it addresses the inherent non-stationary nature of EEG signals, which traditionally complicates the detection of neural activity patterns. Second, AM dynamics features have been shown to be more robust to artifacts in passive BCI applications (e.g., [46]); thus, they may be able to assist with artifact robustness for active BCIs. Third, cortical hemodynamics measured with fNIRS have been found to be correlated with amplitude modulations measured from EEG signals [47], suggesting that amplitude modulation is a good indication of local neural processing. Since multimodal EEG–fNIRS systems have been shown to outperform EEG systems alone [48], the proposed features may be able to capture multiple signal modalities using a single electrode, thus also improving user experience while making BCIs more robust. Lastly, cross-frequency amplitude and phase coherence features have been linked to different cognitive processes; thus, they may further assist with inter- and intra-subject variability [49,50,51,52]. To validate the proposed method, we used the multitask BCI database described in [48] and provide an in-depth discussion of the potential neural processes captured by these new features, providing insights into their importance in the active BCI field.

## 2. Materials and Methods

In this section, we describe the dataset, extracted features, and feature ranking and classification algorithms used.

### 2.1. Experimental Protocol

The present study used the open-source BCI database described in [48]. This dataset investigates the discrimination of distinct neural response patterns associated with seven different mental tasks using features extracted from both EEG and fNIRS modalities. The seven mental tasks include mental rotation (ROT), word generation (WORD), mental subtraction (SUB), mental singing (SING), mental navigation (NAV), motor imagery (MI), and face imagery (FACE). Mental rotation, word generation, and mental subtraction are classified as brainteasers, while mental singing, mental navigation, and motor imagery are classified as dynamic imagery tasks and face imagery is classified as a static imagery task. Table 1 describes each mental task.

Multimodal data were collected from 12 participants who were fluent in English and/or French, had no history of neurological disorders, and had no previous experience with BCIs. The participants consented to participating in the study and monetary compensation was provided after each completed session. All participants agreed with the terms and conditions of the study, which was approved by the INRS Research Ethics Committee.

The data were collected over three recording sessions of 2 to 3 h each, during a period of 3 to 5 weeks. Each session consisted of four sub-sessions, in which each mental task was randomly repeated four times. Each sub-session began and ended with a 30 s baseline period, during which the participants were asked to remain in a neutral mental state and fixate on the cross at the center of their screen. Before each trial, a 3 s countdown screen identified the task to be performed. Once the countdown was over, the participants had to execute the required mental task for a period of 15 s and were instructed to carry out the task as many times as possible during that period. Each trial was followed by a rest period of random duration, sampled from a uniform distribution of between 10 and 15 s, in which participants were asked to continue minimizing movements but were allowed to blink, swallow, etc. The participants were also required to complete a subjective evaluation questionnaire between the second and third sub-session of each session. The stimuli and questionnaire were implemented using Presentation software (Neural Behavioral Systems, USA). More details about the experimental protocol can be found in [48].

EEG data were recorded using a BioSemi ActiveTwo system with 62 electrodes and 4 electrooculography (EOG) electrodes. The experiment used a standard 10-10 system for electrode placement, but without AF7 and AF8, where holders were used for fNIRS probes instead. Only data from nine participants were analyzed as data from Participants 2 and 8 were rejected due to excessive artifacts and higher overall drowsiness levels observed during the recordings. Data from Participant 12 were rejected because the three required sessions were not completed. These participant exclusions matched those suggested in [48] and were replicated here to facilitate comparisons. Moreover, while 60 fNIRS optodes were also included in the dataset, they were not analyzed in this study.

### 2.2. Dataset Pre-Processing

Utilizing the publicly available EEGLAB MATLAB toolbox [53], we first pre-processed the raw EEG signals. Initially, the EEG dataset was re-referenced to the Cz channel, which was later removed. To eliminate electrical grid noise, we applied a notch filter of between 59 and 61 Hz. To tackle high-frequency noise and signal drift arising from electrode impedance variation, a band-pass finite impulse response (FIR) filter, ranging from 0.1 Hz to 50 Hz, was employed.

To facilitate comparisons to the results in [48], channels P8 and O1 were also discarded. Moreover, the first 2 and 14 markers from subject 3 (session 1) and subject 5 (session 1) showed discrepancies with the experimental protocol and were also discarded. We then used the fastICA algorithm [54] to extract independent components. These components were further evaluated using the ADJUST algorithm [55] that is available in the EEGLAB toolbox, allowing us to identify and automatically remove components associated with artifacts. To mitigate the potential impact of lost data, all channels removed in the earlier stages of pre-processing were restored using spherical interpolation. Following this, the EOG channels were removed and the dataset was re-referenced to the average. For detailed examination and replication purposes, the pre-processing script is available at the following GitHub repository: https://github.com/OlivierRS/EEG-Preprocessing-with-ADJUST, accessed on 19 November 2023.

### 2.3. Feature Extraction

#### 2.3.1. Power Spectral Density (Baseline) Features

As a means of comparison to the proposed amplitude modulation features, we utilized power spectral density (PSD) features as the baseline. Initially, the pre-processed EEG signals were segmented into epochs, starting 1 s before the beginning of each mental task and continuing for 15 s thereafter, generating a total epoch duration of 16 s. Subsequently, these epochs were subdivided into non-overlapping 1 s time windows. The PSD was calculated from each of these windows using Welch’s method, employing the MNE toolbox [56], where the power of each frequency was normalized by the sum of the entire power spectrum. PSD features were extracted from several conventional frequency bands, including theta (4–8 Hz), alpha1 (8–10 Hz), alpha2 (10–12 Hz), beta1 (12–21 Hz), beta2 (21–30 Hz), theta to beta (8–30 Hz), and delta to gamma (0–50 Hz), for each EEG electrode. In addition to the PSD features, we also extracted power ratios, notably alpha (8–12 Hz) to beta (12–30 Hz) and theta to beta ratios, for each electrode. Subsequent to the extraction process, all features underwent log-scaling.

#### 2.3.2. Proposed Amplitude Modulation Power Features

The extraction of the amplitude modulation power (AMP) features is presented in Figure 1. The process involved the decomposition of EEG signals into five spectral bands utilizing a filter bank (FB). The bands selected for this study included the conventional delta (1–4 Hz), theta (4–8 Hz), alpha (8–12 Hz), beta (12–30 Hz), and gamma (30–50 Hz), given their established roles in representing neuronal dynamics [57]. We used a zero-phase FIR filter to execute the signal filtering, which consisted of two successive filtering steps in opposing directions on a mirror-padded version of the input signal. The outcome was a series of signals that represented the power dynamics across each frequency band over time. We subsequently obtained the envelope using the absolute value of the Hilbert transform. This envelope was then filtered using the same filter bank, culminating in a set of 5-by-5 modulation signals (shown in the bottom part of the figure) that represented the spectral decomposition of the power dynamics within each specific EEG frequency band, i.e., the measure of the amplitude–amplitude coupling of two EEG frequency bands.

For notation, we refer to these 25 signals as ‘<modulated band>-m<modulant>’, where ‘modulated band’ refers to one of the five bands from the first filtering step and ‘modulant’ pertains to the band applied during the second filtering step. A visual representation of the entire set of the AM time series according to this convention is exhibited in the bottom matrix of Figure 1. However, as per Bedrosian’s theorem [58], not all of these 25 signals could be possible; in fact, only 14 of them could, as low-frequency bands modulated by high-frequency bands were invalid. A more in-depth description of the amplitude modulation-based features can be found in [46,59].

As mentioned previously, while AMP features have been explored for use as biomarkers for neurodegenerative diseases or passive BCIs, they were explored here for active BCI control. One important aspect of active BCIs is being as close to real-time as possible in order to maximize the information transfer rate. However, AMP features have an inherent latency in that envelopes greater than 1 s (i.e., the window sizes typically used for PSD features) are needed. In the past, 8 s windows have been shown to be optimal for biomarker development [59]. Here, we proposed to reduce this to 4 s but with a sliding window of 3 s, suggesting that decisions can be made every second after the initial ‘buffer’ period. This approach yielded 11 distinct feature time ‘frames’ per trial for each frequency band, covering the total trial duration from −1 to 15 s. The features were normalized by the total spectral power at the corresponding time frame.

#### 2.3.3. Phase Circular Correlation of Amplitude Modulated Signals

In addition to analyzing power time dynamics using AMP features, we also extracted features to quantify the connectivity between cortical sites. Although metrics such as phase locked value and magnitude squared coherence have traditionally been used for this purpose, they have been criticized for their vulnerability to coincidental phase synchrony, which can result in the misinterpretation of the results [60]. To address this, we used the phase circular correlation of AM signals (CCORAM) method [60] to examine the connections between different electrode pairs, a choice motivated by the increased robustness of the method, as well as its ability to account for phase co-variance within electrode pairs. To reduce computational demands and avoid excessive dimensionality, we only used a subset of electrodes, specifically F7, F8, T7, T8, C3, C4, P7, P8, O1, and O2. This selection was based on a strategy to maximize the spatial span using a 19-channel montage and minimize the effects of source propagation by excluding neighboring channels [61,87]. With this approach, we calculated the CCORAM for each possible pair within the selected 14 AM bands.

### 2.4. Feature Selection, Classification, and Figures of Merit

Feature selection methods are essential in removing redundant features and avoiding the curse of dimensionality. Methods such as minimum redundancy maximum relevance (mRMR) [62] and recursive feature elimination (RFE) [63] account for interactions among features in their selection process, aiming to minimize redundancy within the selected feature set. Despite their effectiveness, these techniques pose computational challenges, especially in scenarios involving datasets with extensive numbers of features. In contrast, the Fisher linear discriminant (FLD) [64], a filter-based selection method, offers a more practical solution as it evaluates individual features iteratively with significantly lower computational complexity.

For classification, we employed a stratified shuffle split cross-validation approach. In this approach, the aggregated data from all participants were divided, maintaining a 90-10 split for training and testing, respectively, ensuring the proportional representation of both classes in each subset. This procedure was reiterated 100 times, using a different random split each time, to foster a robust estimation of the model’s performance. The hyperparameters of the SVM, with radial basis function kernels, C, and gamma, were optimized in the training set, leveraging a cross-validated grid search. For the figures of merit, we used the average kappa score metric on the predicted labels from the test set for each of the 100 bootstrap trials.

### 2.5. Eigendecomposition-Based Ranking of Binary Classifications

In this study, we examined the interactions between mental tasks via a series of 21 binary classification tasks, aiming to critically assess the influence of amplitude modulation-based features. However, this approach posed a challenge as the insights derived from individual binary classification could only offer a partial understanding of the mental task patterns. To address this, we employed a methodology utilizing eigendecomposition to derive a ranking score, hereafter referred to as the ’prestige score’ based on terminology from the established literature [65]. The detailed process of extracting the prestige score is depicted in Figure 2.

This approach provides insights into the relative importance of mental tasks by considering the entire set of classification performances. The prestige score not only reflects the immediate kappa scores of mental tasks in discriminating against other tasks but also synthesizes the entire set of classification performances represented by kappa scores for task pairs. In particular, when a task is inherently difficult to identify and consequently yields a low discriminative score, having one or two binary classifications with high kappa scores can lead to the overestimation of its true discriminative capacity. With this methodology, we could analytically discern the genuine discriminative scores of the tasks that secured high kappa scores, revealing the relative ease of achieving such scores and thereby justifying the adjustment of these scores to accurately reflect the task’s true discriminative nature.

Figure 2 depicts an illustration of the process of extracting the prestige scores. On the right side of the figure, the step-by-step procedure for applying eigendecomposition to a 7 × 7 interaction matrix is demonstrated. This matrix was constructed by incorporating the 21 kappa or Fisher score measurements in its upper triangle, each element of which represented a measurement from a pair of tasks. The analytical process yielded the prestige score, associated with the eigenvector with the highest eigenvalue. Utilizing the Perron–Frobenius theorem, these prestige scores were identified as normalized positive vectors that revealed the latent influences that each class held as dictated by the chosen metric [66]. To maintain the relative magnitudes of the interactions captured in the different dataset folds during data amalgamation, each prestige score was scaled by its corresponding eigenvalue. This approach avoided the empirical analysis that is traditionally conducted on 7 × 7 interaction matrices, allowing for a more robust and succinct interpretation of the influences of mental tasks through 7-element vectors. Consequently, plotting the feature distribution of individual mental tasks became a more streamlined alternative to the conventional method of illustrating all 21 pairs of mental task combinations. Subsequently, the left side of Figure 2 clarifies the iterative procedure employed to amalgamate the prestige scores.

## 3. Experimental Results

In this section, we describe the experimental results in terms of the impact of the proposed features, the ranking of the mental tasks, and a discussion on the top-selected features.

### 3.1. Estimation of Optimal Feature Set Size

To estimate the most suitable balance between reducing overfitting risk and ensuring high classifier accuracy, we proceeded to identify the optimal number of features by systematically testing various feature set sizes. Figure 3 depicts the relationship between the quantity of the top-selected features, ranked using the Fisher linear discriminant scores, and the performance of a vanilla SVM classifier. The x-axis represents the number of selected features and the y-axis represents the grand average kappa score derived from all 21 task pair classifications. In total, three combinations of feature types were tested and the model performance using PSD, AMP+PSD, and AMP+CCORAM+PSD are shown in yellow, orange, and blue, respectively. We used the kappa scores to analyze the curve and determine the exact number of features for when the accuracy started to plateau. As can be seen, this occurred after 2000 features for all feature type combinations. Above this threshold, the performance gain was negligible and potential overfitting issues could appear. Henceforth, only experiments with classifiers trained using this number of top features will be reported.

Following feature selection, we conducted an in-depth analysis of the distribution of all selected features, categorizing them by type and analyzing them within each top feature subset based on both appearance frequency and FLD score. The AMP features were predominant, representing 74.14% of the selected pool with a cumulative score of 70.02%. PSD features followed, accounting for 22.72% with a cumulative score of 27.11%. The CCORAM features were the least represented at 3.14%, contributing 2.86% to the cumulative score.

### 3.2. Impact of Proposed Features on Classification Performance

Figure 4 shows the kappa scores obtained for each of the 21 possible task pairs for PSD features alone (orange), PSD features combined with the proposed AMP features (red), and all three feature sets together (blue). As can be seen, the incorporation of AMP features significantly enhanced performance across 17 of the 21 task pairs. For the remaining four task pairs, the accuracy still increased in three, just not significantly (i.e., ROT-WORD, NAV-WORD, and FACE-SING), while the accuracy actually dropped relative to using PSD features alone in one (ROT-SUB). Task pairs associated with mental activities, such as SUB, WORD, FACE, and SING, generally showed negligible enhancements. Notably, the MI-SING and FACE-SING pairs registered the lowest kappa values. These tasks, which are intricately linked to cognitive functions, such as working memory, long-term affective retrieval, and both auditory and visual memory processing, could inherently possess greater inter-subject variability due to their abstract nature, while factors such as individual cognitive approach, mental state, and problem-solving strategy could influence the outcome, leading to subjective disparities.

Overall, the average kappa scores across all 21 task pairs were 0.6221, 0.6237, and 0.5761 for the AMP+CCORAM+PSD, AMP+PSD, and PSD models, respectively. These findings suggest that in the majority of cases, AMP features capture exclusive information that is not obtained through PSD alone.

### 3.3. Ranking of Mental Task Kappa Scores

By employing the eigendecomposition ranking technique, we estimated the discriminative power of the individual mental tasks. Figure 5 illustrates the estimated individual kappa scores for each of the seven mental tasks, which were derived from the averaged prestige score vectors across the 100 dataset folds. These vectors were procured through the ranking method outlined previously, employing the kappa score interaction matrix. In essence, each iteration of the classification pipeline produced a single interaction matrix, from which a prestige score was extracted. This procedure was repeated across all folds to ultimately average the prestige scores, thereby providing an estimate of the ”true” kappa score for each mental task. As a result, mental tasks with higher relative scores were suggestive of a consistently superior discriminating pattern quality, revealing exclusive patterns that distinctly characterized such tasks. The outcomes of this analysis are shown in Figure 5, where the discriminatory power of each task is displayed in decreasing order of prestige score.

As can be seen, the rotation (ROT) task achieved the highest score. Interestingly, in [48], this task was shown to be preferred amongst the participants, despite its demanding nature. This capacity for differentiation could be attributed to its primarily visual nature, which reduces the subjectivity typically present in tasks involving complex processing, such as memory or emotion, thereby exhibiting more discernable neuronal patterns. Conversely, our findings showed that the FACE and SING tasks were the least accurately identified, a result that was in alignment with feedback given by subjects in [48], suggesting that these were their least favorite tasks.

The navigation (NAV) and subtraction (SUB) tasks placed second and third in this analysis, both of which were liked by subjects in [48]. Interestingly, while the motor imagery (MI) task demonstrated a similar kappa score to the FACE and WORD tasks in our study, its use is widespread within the active BCI community. It has been hypothesized that MI-induced patterns may be overshadowed when paired with brain teaser tasks. This could be observed in task pairs such as FACE–MI and SING–MI. The intricate dynamics of MI, involving both promising classifications in certain configurations and limitations in others, underscores a complex landscape that necessitates a careful and balanced consideration when selecting tasks for BCI applications.

### 3.4. Mental Task Feature Analysis

In a similar vein to the analysis performed in the previous section and depicted by Figure 2, we re-conducted this analysis with the goal of understanding which feature patterns most efficiently captured neural patterns for different tasks. We started by constructing a 7 × 7 matrix for each feature. Each cell in these matrices corresponded to the FLD score for a pair of mental tasks when differentiated using that particular feature. Consequently, each matrix provided a snapshot of the discriminative power of a single feature across different task pairs. From these matrices, prestige scores were derived for each feature, reflecting its distributed discriminative power across the mental tasks.

This analysis was repeated for all features and then aggregated per feature type group, i.e., AMP, CCORAM, or PSD. Figure 6 summarizes our findings. As can be seen, the AMP features were the most efficient in capturing discerning patterns for each individual mental task, followed by the traditional PSD features. Particularly in the ROT task, AMP demonstrated a marked effectiveness over PSD, whereas for the SUB task, both AMP and PSD showed similar discriminative power. Meanwhile, the contribution of the CCORAM feature type was notably lower, showing no clear leaning towards any specific task, indicating its limited utility in this context. It is hoped that these findings will provide insights for researchers developing active BCIs regarding what features to use depending on the mental task in question.

### 3.5. Multidimensional Analysis of Relevant Features

For each individual feature, which were associated with unique combinations of channels, bands, and feature types, we computed a distinct prestige score, representing their ranking across mental tasks. Utilizing these scores, we constructed topoplots to visually represent the distribution of feature rankings across the EEG scalp. For the AMP feature type, the corresponding topoplots for each band are illustrated in Figure 7, Figure 8, Figure 9, Figure 10 and Figure 11. Similarly, for the PSD feature type, Figure 12, Figure 13 and Figure 14 display the topoplots of the prestige score distributions for each band.

## 4. Discussion

The proposed amplitude modulation features are postulated to mirror core biological mechanisms within the brain that are potentially engaged in processing stimuli and directing behavioral responses [42,67,68,69]. Numerous studies have suggested that amplitude modulation patterns in EEG signals could be indicative of mechanisms for controlled inhibition [69,70,71,72] and cognitive process integration [73,74,75], and correlate with energy consumption in cortical tissues [76]. Based on our experiments, such features offer a supplemental layer of information that could enrich traditional PSD features in classifying mental states and tasks, thus improving the robustness of active BCI systems. In the following subsections, we postulate the underpinnings of these new features and their complementary roles to PSDs.

### 4.1. Beta Band Analysis

The beta frequency sub-band, which serves numerous roles depending on its location and context in the brain [73,77], is dominantly involved in long-term memory and stimuli processes [78,79]. It facilitates the binding of temporally separated information into meaningful entities and aids in maintaining task-specific cognitive processes over extended periods [73]. Figure 14 illustrates topoplots that depict the spatial distribution of the prestige scores for relevant PSD features associated with the beta band. A comparative examination of the topoplots for the FACE and ROT tasks revealed analogous patterns, with both demonstrating bilateral frontotemporal patterns in the beta band.

A similar bilateral frontal pattern is seen in the beta–mbeta band in the AMP topoplots in Figure 10. Additionally, the AMP topoplots unveiled noteworthy patterns in the parieto-occipital-temporal region for the beta–mdelta and beta–mtheta bands. Beta activity modulated by the lower delta and theta bands was disseminated across the scalp, while the distribution of faster beta dynamics, specifically beta–malpha and beta–mbeta, was inclined towards the frontal area. The observation of slow temporal beta activity dynamics in the occipital-parietal area could suggest top-down control via the slow wave dynamics of specialized visual processing brain structures [73,80,81,82]. This activity is congruent with the visual nature of the FACE task. Conversely, fast beta activity dynamics were localized in the frontal area, implying the potential association with long-term memory and recall processes. It is well established that slow waves play a pivotal role in facilitating long-distance communication between distinct brain structures, with the interactions between the beta and theta bands contributing to the binding of neuronal information across both time and space [73,83].

In both the motor imagery (MI) and navigation (NAV) tasks, there was a notable presence of activity in the motor cortex, emphasizing their shared reliance on motor functions. For the MI task, the literature has typically reported amplitude changes in the mu band (7.5 Hz to 12.5 Hz) and beta band oscillations in the motor cortex [84,85], although this is not evidenced in the PSD topoplots presented in Figure 14. Nevertheless, this typical MI beta activity can be observed in the AMP topoplots in Figure 10. Here, beta activity in the MI task could be delineated into beta–mtheta and beta–mbeta AM bands, suggesting a role for beta in maintaining long-term motor action.

For the NAV task, a slight similarity was discerned between the distributions of the beta1 PSD and corresponding AMP topoplots. This resemblance unveiled insights into the multifarious nature of beta band activity, hinting at processes such as the recall of familiar memories and the long-term binding of information. In particular, the beta–mdelta and beta–mbeta bands in the AMP topoplot exhibited relevant bilateral frontotemporal and occipital patterns. We hypothesized that beta–mdelta activity in frontotemporal areas was indicative of processes related to recalling familiar memories, possibly due to the involvement of the limbic system [73,81,83,86,87]. This system is a well-known generator of delta band oscillations, with delta band activity from the medial temporal cortex being notably associated with assessing the familiarity of sensory stimuli [76,88,89].

Further, beta–mbeta activity may be associated with the long-term temporal binding of information, a crucial aspect considering that the NAV task involved navigating through familiar terrain over extended durations. This band, depicting the fast temporal dynamics of beta oscillations in the prefrontal area, could reflect communication from the limbic system to the cortex, a mechanism that is integral to working and long-term memory processes [73,82,83]. In addition, beta–mtheta activity revealed an exclusive pattern in the left cortical hemisphere, potentially related to memory restitution.

In both the ROT and SING tasks, a general agreement between AMP and PSD was observed, with the presence of beta–mbeta activity in the motor cortex aligning with the necessity for advanced visuomotor functions in ROT and the long-term maintenance of specific processes in SING [73,81]. These observations collectively highlighted the consistent binding role of beta–mbeta activity across varied tasks.

In the PSD topoplots in Figure 14, relevant activity in the beta1 band was observed in the parietal and motor cortices during the WORD task. This pattern can be similarly observed in the AMP topoplots in Figure 10, within the beta–mdelta, beta–mtheta, and beta–mbeta bands. Notably, there was a diminished contribution from the anterior frontal region in the case of the beta–malpha band.

Following these observations, the AMP topoplots illustrated in Figure 10 underscored the significance of the motor cortex during the WORD task. This suggests the pivotal roles of motor functions in linguistic tasks [73,90]. The presence of the beta–mbeta and beta–mtheta bands further accentuated the potential of fast beta band dynamics in binding large neuronal assemblies [73,80,91].

### 4.2. Theta Band Analysis

As shown in Figure 12, the FACE task presented a relevant feature distribution in the occipital-parietal region and a partial presence in the frontal cortex. This pattern could be divided into two complementary patterns observed in the theta–mdelta and theta–mtheta bands in Figure 8. While the theta–mdelta band clearly represented most of the relevant activity located in the prefrontal, motor, and occipital cortices, the theta–mtheta band exhibited precise relevant activity in the occipital-parietal area. The theta band is known to be involved in memory retrieval [76,77,78,82,86,87,92] and the process of visual information encoding. This activity aligned with the requirements of the FACE task, which was a visual task necessitating memory retrieval.

For the MI task, the slow temporal dynamics of the theta band, represented by theta–mdelta, suggested modulated attention across time by the PFC. In contrast, faster theta power changes, represented by the theta–mtheta band, seemed mainly to contribute to the decoding of recalling visual information. In the MI task, the patterns observed across PSD and AMP, particularly in theta–mtheta band, supported the hypothesis of sensory feedback processing, providing a nuanced understanding of the reactivation of specialized sensory brain structures.

In the NAV task, both the PSD and AMP topoplots in Figure 8 and Figure 12, respectively, showcase parietal and prefrontal activity, hinting at the activation of visual processing regions and working memory involvement, with theta oscillations likely reflecting a decoding mechanism for previously learned information [75,80]. Similarly, the ROT task presented theta patterns in the occipital-parietal regions that are associated with visual stimuli encoding, while frontal patterns in theta–mdelta band could be interpreted as reflecting problem-solving processes.

Despite the absence of prominent patterns in PSD for the SING task, the theta–mtheta activity in AMP revealed the involvement of the frontal and parietal regions, potentially indicating memorized information retrieval, maintaining the consistent hypothetical role of theta’s involvement in information encoding [76,78,93].

Regarding the WORD task, the AMP topoplots in Figure 8 provide nuanced insights into the theta band’s temporal dynamics and reveal a left prefrontal cortex lateralization, aligning with the literature on hemisphere association with linguistic functions. Further, the importance of the motor cortex, as highlighted in the AMP topoplots, aligned with the literature regarding its involvement in linguistic tasks [94].

### 4.3. Alpha Band Analysis

By examining the FACE task in Figure 13, it can be seen that both the alpha1 and alpha2 bands exhibited relevant occipital-parietal spatial distributions, as well as left frontal distribution. In the AMP topoplots presented in Figure 9, such patterns are also identifiable; however, the alpha–mdelta band showed diffused relevant activity across the entire scalp, while the alpha–mtheta band distribution was focused in the frontal and left occipital areas. These findings suggest two levels of speed in the alpha power temporal dynamics: slow variation in alpha activity seemed to be related to global scope function, while faster temporal dynamics seemed to be related to specific functions, such as executive and visual processing. Alpha oscillations, known for their relation with inhibition [70,81,95], show modulation by slow waves in the delta band, indicating the possible involvement of the limbic system as a modulated attention process. We could hypothesize that remembering a familiar face requires the sustained inhibition of new stimuli processing, as well as coordination between visually specialized brain structures (occipital) and specialized executive/emotion-related brain structures (frontal) [95].

Regarding the MI task, the mu band (7.5 to 12.5 Hz) is typically considered to be a relevant oscillation frequency within the motor cortex during motor imagery [96,97]. This pattern was also discernible in our feature analysis of the PSD topoplots in Figure 13, where we noted relevant activity for both the alpha1 (8 to 10 Hz) and alpha2 (10 to 12 Hz) bands in the motor cortex, coupled with observations of parietal-occipital activity for both bands. Upon contrasting this observed activity with that in the AMP topoplots, similar distributions of the relevant features were evident in the alpha–mdelta and alpha–mtheta bands. Specifically, the topoplot of alpha–mdelta band in Figure 9 illustrates pronounced and widespread activity in the motor cortex, suggesting that mu activity becomes more discernible when considering the slow temporal dynamics of alpha oscillations. We further hypothesized that the MI task necessitated the partial deactivation of the motor region as the movement was imagined and not executed, explaining the presence of slow modulated inhibition in the motor area. Furthermore, the alpha–mdelta band exhibited highly relevant activity in the parietal-occipital regions, potentially indicating the inhibition of visual stimuli [77], analogous to observations made in the FACE task. The alpha–mtheta band displayed a subdued pattern in the parietal-occipital cortex, with the distribution being more centralized in the somatosensory cortex. This activity was noteworthy as it suggested the presence of some form of virtual sensory feedback accompanying the imagined motor activity.

For the ROT task, the PSD topoplots depicted in Figure 13 exhibit consistent patterns for the alpha1 and alpha2 bands, showcasing activity in the bilateral frontal and right occipital regions. Delving deeper into the AMP topoplots, as seen in Figure 9, a clear distinction arises between two brain areas: the alpha–mdelta band was concentrated in the right occipital area, while alpha–mtheta band was prominently active in the bilateral frontal regions. This differentiation implied that the alpha–mdelta band was notably associated with controlled attention in the visual cortex, a nuanced insight provided by the comprehensive temporal analysis inherent to AMP features. This discerned alpha activity in both frontal and occipital areas potentially reflected a dynamic shift in attention and modulated inhibition across time. This interpretation aligned coherently with the demands of the ROT task, which necessitated the processing of visual stimuli coupled with logical evaluation.

For the SING task, the alpha–mtheta band was notably relevant in the prefrontal cortex and left parietal cortex, as evidenced more clearly by AMP than PSD. The alpha–mdelta band presented a comparable, albeit more diffuse, cortical-frontal activity. Unlike the other tasks, SING did not primarily involve visual processing, explaining the absence of any significant occipital activity. However, the task could necessitate imagined visual constructs or advanced sensorial representation, implicating the somatosensory, motor, and prefrontal cortices. The pronounced presence of alpha–mdelta activity in the prefrontal cortex could signify the greater influence of sympathetic structures during song recollection.

Intriguingly, the SUB task exhibited distribution patterns analogous to those of the SING task. This similarity was characterized by pronounced prefrontal activity in both the alpha–mdelta and alpha–mtheta bands, along with a bilateral occipital-parietal pattern. This pattern is observable in the alpha1 and alpha2 bands in the PSD topoplots represented in Figure 13 and the alpha–mdelta and alpha–mtheta bands in the AMP topoplots depicted in Figure 9. These observed distributions suggested the potential reliance on working memory and visual cues in the SUB task [76,90,98]. Lastly, the AMP topoplots for the WORD task revealed a left asymmetry in the cortical distribution of relevant features, a characteristic only present in the alpha2 band in the PSD topoplots. This asymmetry aligned with reports in the existing scientific literature [99].

### 4.4. Delta Band Analysis

As shown in Figure 7, the delta band exhibited a uniform distribution of relevant features across the scalp in all tasks, illustrating its role in modulating various cognitive processes throughout the brain. Notably, in the FACE and ROT tasks, the increased relevance of delta activity in the bilateral frontotemporal and parietal-occipital regions was observed, emphasizing its pivotal role in modulating attention [71,72] and facilitating the integration and coordination of sensory information [79]. This temporal synchronization is especially crucial in tasks requiring coherent communication among different brain regions for structured cognitive task execution. In the MI, SING, and SUB tasks, a subtle yet noticeable relevance in the parietal-occipital regions underscored the delta band’s versatility in contributing to functions like motor imagery, sensory processing, and working memory. The insights derived from analyzing the delta band AMP topoplots across tasks (Figure 7) underscored the delta band’s fundamental role in attention modulation, the temporal binding of neurological activity, and sustaining cognitive processing.

### 4.5. Gamma Band Analysis

As can be seen in Figure 11, for the FACE task, the relevant gamma band activity in the somatosensory area and bilateral frontotemporal regions emphasized its role in the processing of sensory information associated with recalling previously learned stimuli. The rapid oscillations of the gamma band were indicative of localized neuronal activity [70,73,80], making neurons more receptive to specific stimuli and thereby aiding in the retrieval of specific visual information [76,77,88,93]. In the MI task, the prominence of the gamma band in the motor and premotor cortices was consistent with the localized activation of neuronal structures associated with imagined motor activity. This suggests that the gamma band is closely linked with tasks that involve motor planning without actual movement. In the ROT task, all modulated gamma bands depicted in Figure 11 exhibited consistent distributions of relevant activity across a variety of cortical areas. The observed patterns of activity aligned with the multifaceted demands of the ROT task, which necessitated the concurrent engagement of abstract visual representation, advanced sensory processing, working memory, and executive functions. In both the SING and SUB tasks, the primary patterns in the somatosensory and parietal cortices marked the gamma band’s significance in sensory processing, which is required in tasks in which sensorial patterns are reactivated, such as mental singing. Lastly, the WORD task displayed lateral patterns in the gamma–mdelta and gamma–mbeta bands in the left parietal cortex, with a shift towards the right parietal area in the gamma–mtheta, gamma–malpha, and gamma–mgamma bands. Such hemispheric specialization echoed with the previously observed patterns in the theta and alpha bands and further aligned with the well-established concept of hemisphere dominance in language processing [99].

### 4.6. Performance Interpretation

In light of the distinctive patterns observed across the different bands and mental tasks, we propose additional interpretations for some of the performance results. For the ROT task, unique distribution patterns were apparent in both the alpha and beta bands, depending on the temporal dynamics, alongside noticeable activity in the right motor cortex in the beta–mbeta band. These patterns, revealed by the AMP topoplots, offered additional insights beyond what PSDs could provide, thereby potentially explaining the clear differences in the accumulated prestige scores observed in Figure 6.

When comparing the MI task to the SING task, several similarities emerged, including the involvement of the alpha band in the motor cortex and parietal-occipital regions and the presence of beta band activity in the motor and sensorimotor cortices, as well as frontotemporal areas. These similarities could account for the low kappa score observed for the MI-SING mental task pair in Figure 4. Lastly, the SUB task demonstrated clear similarities in both the PSD and AMP topoplots for the alpha and beta bands and comparable occipital activity in the theta band. These similarities in the distributions of both feature types could account for the small differences in prestige scores observed in Figure 6.

In conclusion, distinguishing between the band amplitude dynamics in slow and fast components significantly enhanced our understanding of the roles of such bands. This was particularly crucial as slow oscillations are typically associated with long-distance communication between distinct brain regions, while fast oscillations correspond to intensive neurological activity. While our analysis provided an initial step towards understanding the neural oscillations and regional distributions underlying different tasks, a complete picture of brain dynamics is yet to emerge. Further investigations into functional connectivity and synchronization between different brain areas during tasks will provide more insights into the complex brain networks involved in mental processes.

Overall, our findings highlight the potential of AM features as promising tools for uncovering task-related changes across different frequencies and brain regions. By reflecting fundamental neurological mechanisms, such as inter-areal communication and top-down control, AM features could potentially enrich our understanding of the complexities of brain activity. These findings underscore the need to incorporate and investigate AM features in future BCI research.

### 4.7. Limitations

This work was not without its limitations. First, the number of participants, while in line with both clinical [37,100] and non-clinical BCI studies [101,102,103], was limited and future work could explore larger population sizes. Moreover, the proposed classification tasks relied on discriminating between two different mental tasks. In the future, a third resting class could also be used. Additionally, while the present study was interested in gauging the benefits of the proposed features and their existence across participants and session dates, the data from all participants were first combined and then partitioned into training and test sets. To further improve classifier accuracy, future work could explore a more common per-subject analysis where classifiers are fine-tuned to each individual user [104] or use data from certain days for training and those from other days for testing [105].

The extraction of AMP and CCORAM features in our study employed a 4 s time interval, introducing some latency. While the impact was minimized with a sliding window with 3 s overlap, the initial latency was a limiting factor compared to, e.g., PSD-based measures, which can rely on windows of 1 s and overlaps of milliseconds. As such, active BCIs based on the proposed features may have lower information transfer rates than other methods. While this may not be an issue for many applications, including neurorehabilitation, future studies could investigate the use of shorter epoch sizes to optimize the balance between latency and accuracy.

Moreover, while using CCORAM provided a more robust estimation of synchronized regions relative to the traditional coherence and phase locked value metrics, it did not reveal the direction of information flow. This information, in turn, could lead to additional insights, such as differentiation between motor control and sensory feedback information [106]. Additionally, phase-to-phase metrics are believed to measure cross-frequency interactions in the brain more accurately compared to phase–amplitude coupling-related features [74]; thus, future work could also explore such measures. Regarding frequency bands, here, we relied on the five conventional bands to allow for comparisons to previous works. However, some active BCIs may achieve improved performance with other bands (e.g., the mu band for MI-based BCIs). Future work could also explore the benefits of using alternative frequency band representations.

Lastly, we chose to use Fisher linear discriminant-based feature selection due to its computational effectiveness and ease-of-use for binary tasks. However, such simple selection methods do not remove potentially redundant features, which could be the cause of the small fluctuations seen in Figure 3. In the future, other more advanced feature selection algorithms, such as the mRMR algorithm [62], may provide a more concise feature pool for analysis.

## 5. Conclusions

In this paper, we proposed a new feature set for active BCIs based on the amplitude modulation dynamics of different EEG sub-bands. Via extensive experimentation, we showed the benefits of the proposed features, as well as their complementarity to conventional power spectral features. An in-depth discussion was provided to explore the complex cognitive mechanisms being measured by the new features and conjecture their roles in improving BCI accuracy for different mental tasks. As for future work, we wish to explore the use of the proposed features for adaptive BCIs, where the mental states of users can be tracked in real time (e.g., fatigue, frustration) to adjust active BCI classification. Such cognitive states are known to affect active BCI accuracy and may be a major limiting factor in BCI use in neurorehabilitation applications [14,107,108].

## Figures and Tables

**Figure 1 sensors-23-09352-f001:**
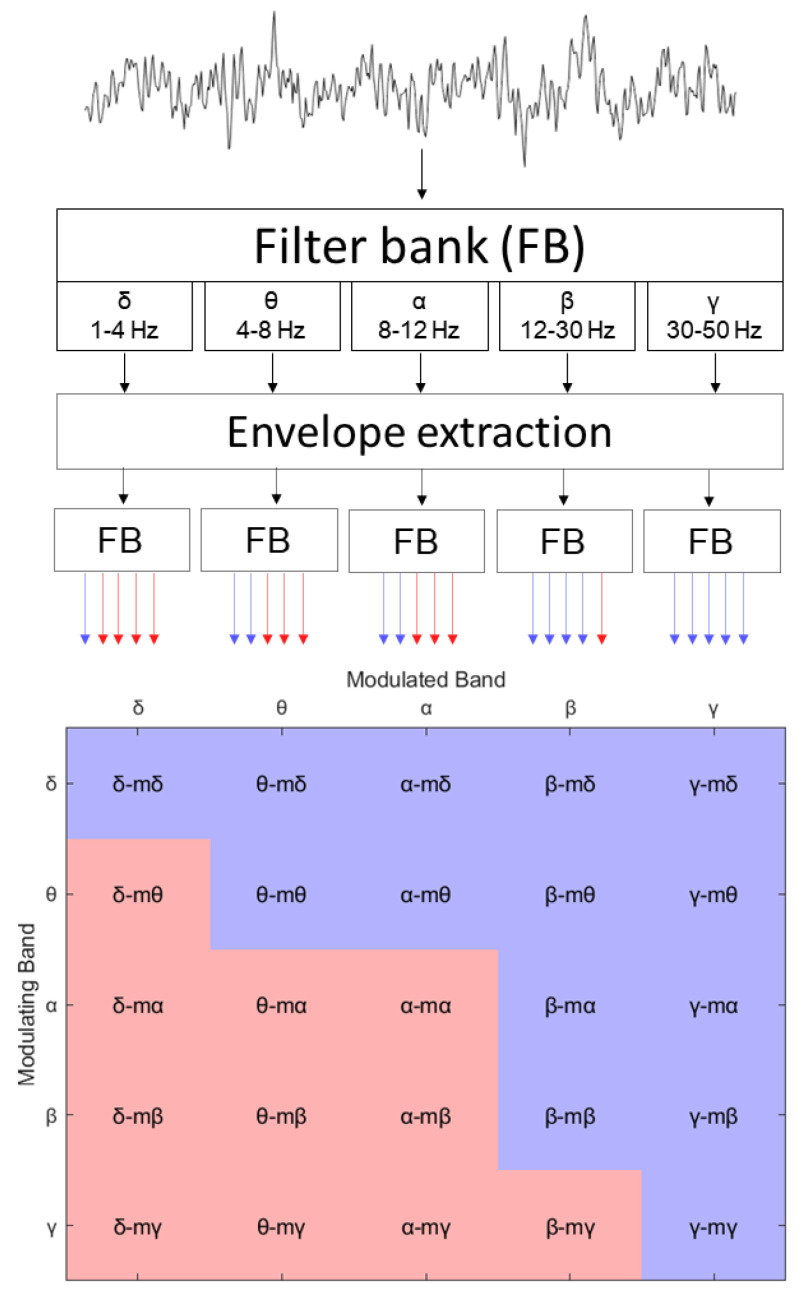
Flow chart illustrating the procedure for amplitude modulation time series extraction from EEG signals. In the top left corner, the straight line denotes the original and band-filtered EEG signal. The dashed lines and adjoining blocks represent the envelope extraction and processing stages. Based on Bedrosian’s theorem, the bottom left matrix differentiates between valid (blue) and invalid (red) amplitude modulation time series.

**Figure 2 sensors-23-09352-f002:**
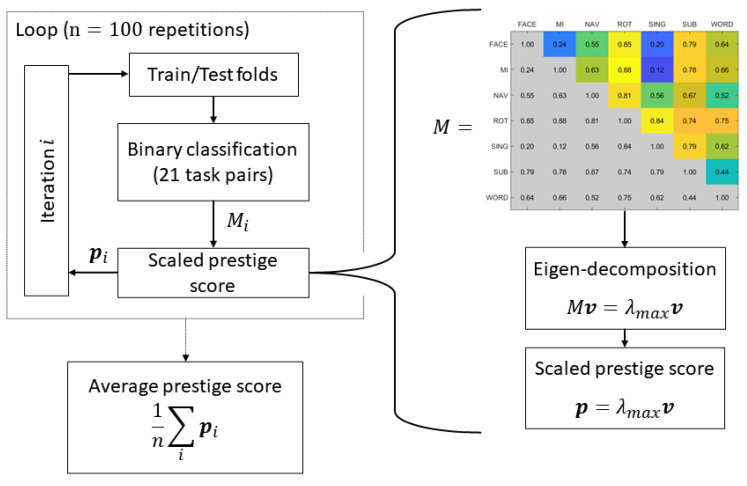
Flow chart illustrating the computation of the scaled prestige scores, depicting how the 21 kappa score measurements were generated in each iteration and arranged in the 7 × 7 interaction matrix ‘M’, followed by the eigendecomposition and final averaging of the prestige scores. Kappa scores are used as an illustrative example.

**Figure 3 sensors-23-09352-f003:**
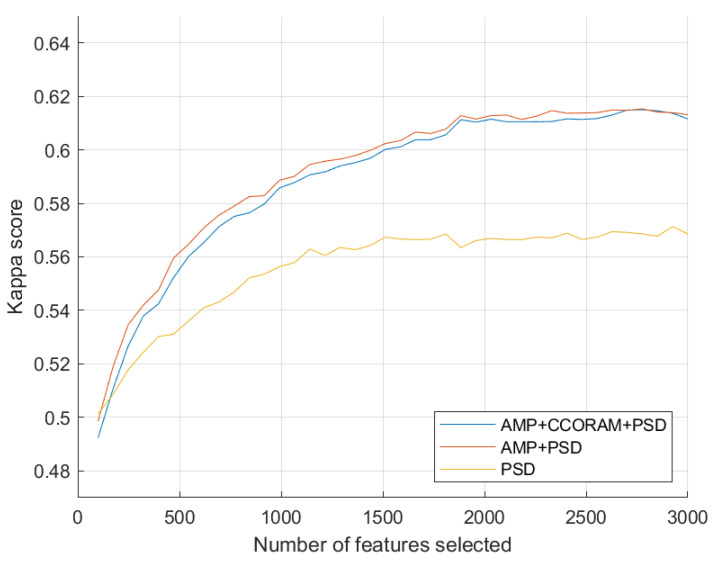
Grand average of task pair kappa scores versus the number of features. The top 2000 features were chosen for subsequent experiments.

**Figure 4 sensors-23-09352-f004:**
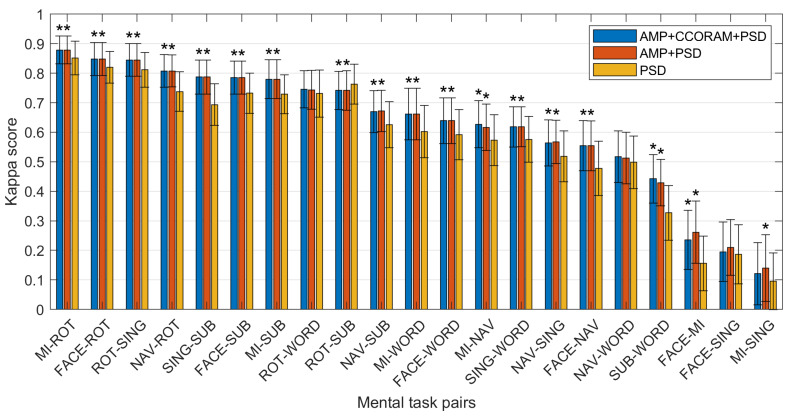
Comparison of the kappa scores of the AMP+CCORAM+PSD (blue), AMP+PSD (red), and PSD (orange) models, derived from the selection of the 2000 optimal features. Columns AMP+CCORAM+PSD and/or AMP+PSD with a symbol * indicate a significant difference relative to PSD results, with a significance level alpha of 0.05.

**Figure 5 sensors-23-09352-f005:**
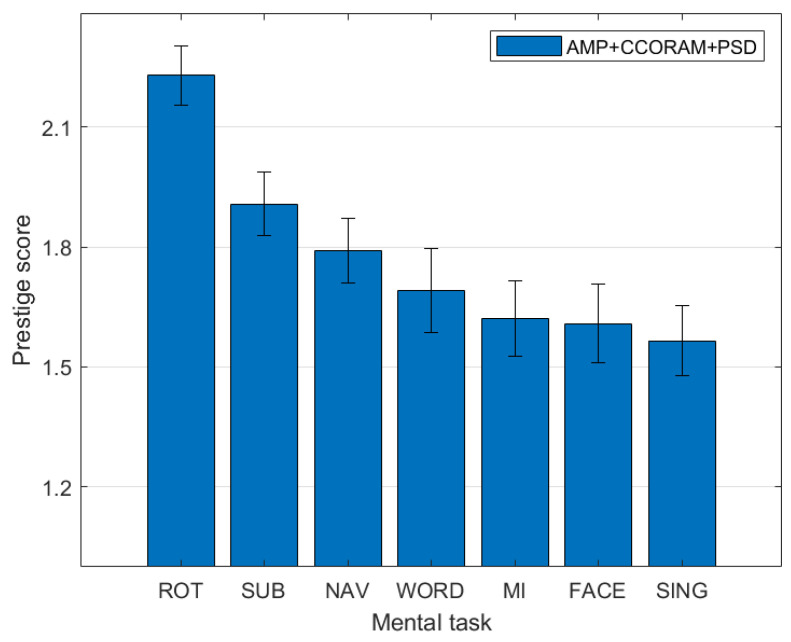
Ranking of individual mental tasks using PSD+AMP+CCORAM features, derived from pairwise classification kappa scores. A higher rank indicates a more distinct pattern, facilitating the model’s ability to discriminate one mental task from others.

**Figure 6 sensors-23-09352-f006:**
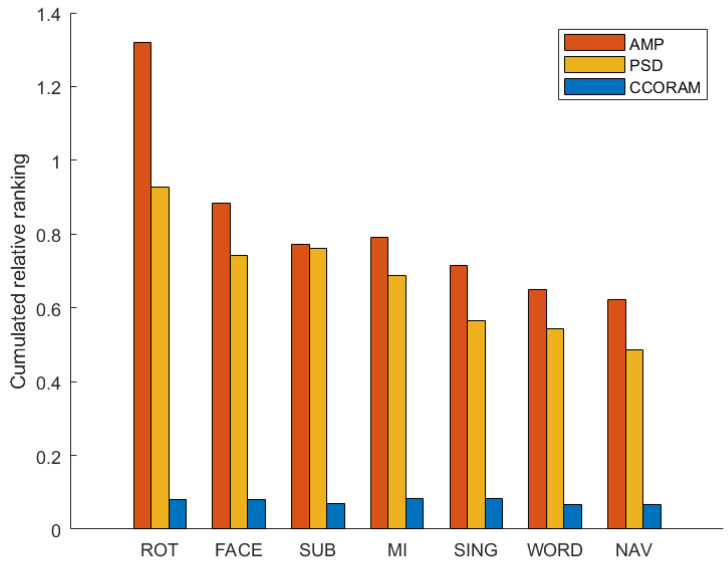
Average Fisher-based prestige scores of mental tasks per feature type (AMP, PSD, and CCORAM).

**Figure 7 sensors-23-09352-f007:**
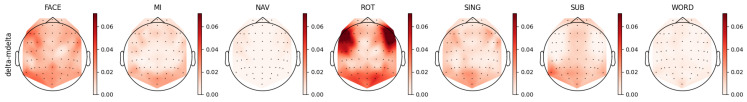
Ranking distribution of AMP features related to the delta band.

**Figure 8 sensors-23-09352-f008:**
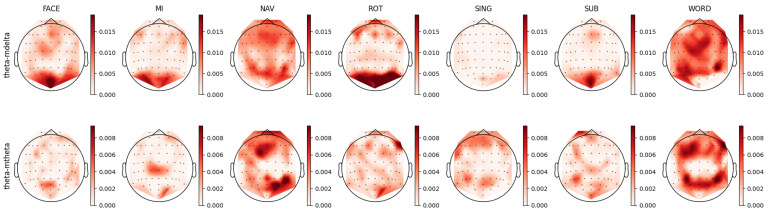
Ranking distribution of AMP features related to the theta band.

**Figure 9 sensors-23-09352-f009:**
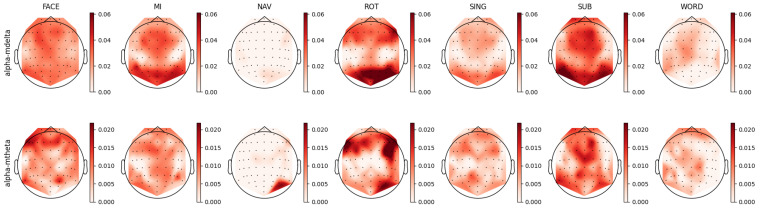
Ranking distribution of AMP features related to the alpha band.

**Figure 10 sensors-23-09352-f010:**
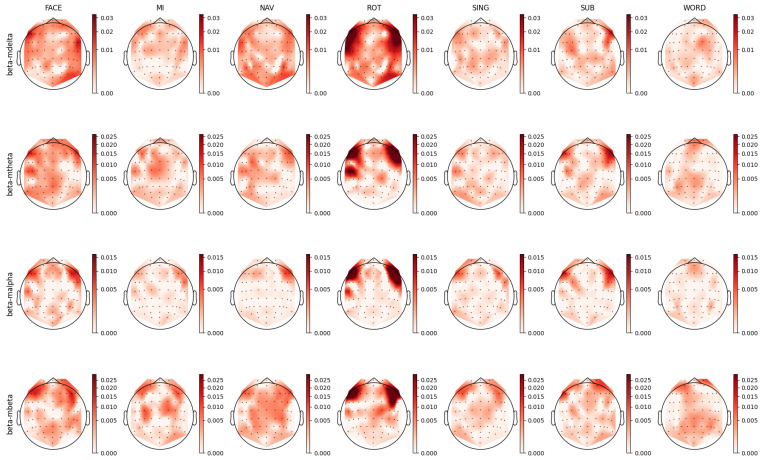
Ranking distribution of AMP features related to the beta band.

**Figure 11 sensors-23-09352-f011:**
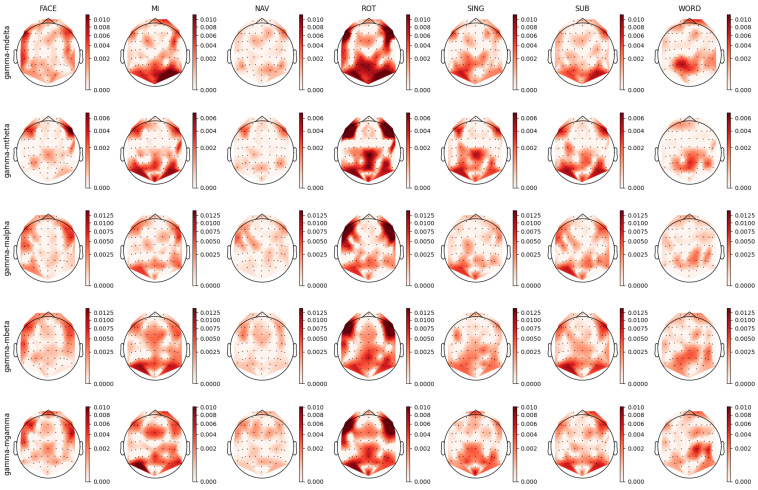
Ranking distribution of AMP features related to the gamma band.

**Figure 12 sensors-23-09352-f012:**
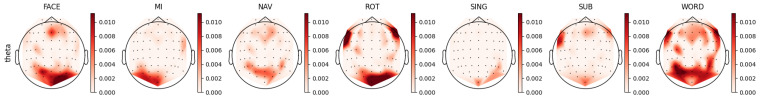
Ranking distribution of PSD features for the theta band.

**Figure 13 sensors-23-09352-f013:**
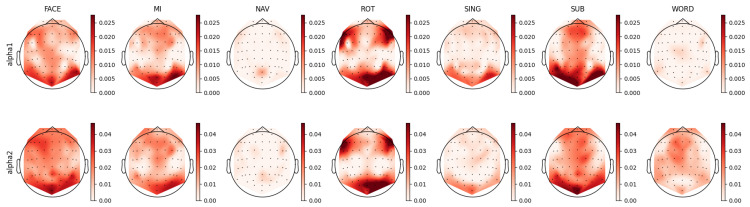
Ranking distribution of PSD features for the alpha band.

**Figure 14 sensors-23-09352-f014:**
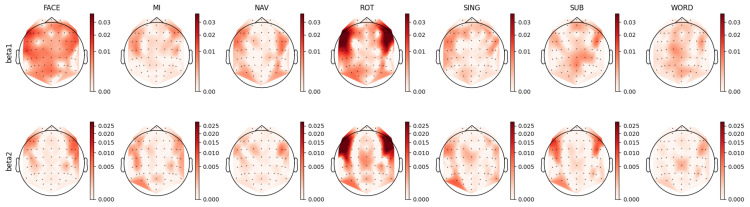
Ranking distribution of PSD features for the beta band.

**Table 1 sensors-23-09352-t001:** Short description of each mental task.

Mental Task	Task Description
Mental Rotation (ROT)	Participants had to imagine the 3D rotation of two objects and determine whether the objects were identical
Word Generation (WORD)	A letter was presented randomly and the participants needed to find as many words as possible
	beginning with this letter
Subtraction (SUB)	Participants had to execute the mental subtraction of 1 to 2 digit numbers from a 3 digit number
Singing (SING)	Participants had to choose a song and then mentally sing it while paying attention to
	the emotions that they felt
Navigation (NAV)	Participants had to imagine walking from one room to another in their past or current residence
Motor Imagery (MI)	Participants had to imagine moving their fingers
Face Imagery (FACE)	Participants had to remember the face of a friend

## Data Availability

Data was collected in 2014, as described in [48]. Data can be made available upon request to T.H.F.

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
