# Peer review of "EEG Amplitude Modulation Analysis across Mental Tasks: Towards Improved Active BCIs"

_sensors, 2023, doi:10.3390/s23239352_

Round 1

Reviewer 1 Report

Comments and Suggestions for Authors

The study aims to improve active BCI robustness by incorporating a new signal representation that allows for the measurement of amplitude modulation (AM) dynamics and cross-frequency coupling from electroencephalography (EEG) signals. The study proposes the use of new features based on the EEG amplitude modulation dynamics. The study observed that addition of the proposed features significantly improved classifier performance relative to using only power spectral features. The findings are significant as they highlight the potential of AM features as a promising tool for uncovering task-related changes across different frequencies and brain regions.

Overall, the manuscript is well written with clear and concise language. However, the manuscript should be further enhances based on the following recommendations:

·       Please add more results into the abstract

·       The literature review appears to focus primarily on technical studies related to BCI and EEG analysis. Incorporating a wider range of sources, including foundational neuroscience literature and research on the psychological aspects of the mental tasks employed, could provide richer context and help readers understand the study's implications.

·       The justification for specific methodological choices, such as the use of certain frequency bands or the Fisher linear discriminant score for feature selection, isn't thoroughly discussed. The paper could strengthen its position by explaining why these methods were chosen over alternatives and discussing the potential impact of these decisions on the results.

·       The study is based on a small sample size of only nine subjects. I acknowledge that the authors have addressed this issue in lines 549-550 in the limitations section. However, the references cited for justification does not involve clinical experiments. To this end, the method and limitation sections could be improved, by adding some recent MI studies that have used similar sample sizes in a clinical setting, and have successfully addressed these concerns; please refer the following reference:

·       Castiblanco Jimenez, I. A., Gomez Acevedo, J. S., Olivetti, E. C., Marcolin, F., Ulrich, L., Moos, S., & Vezzetti, E. (2022). User Engagement Comparison between Advergames and Traditional Advertising Using EEG: Does the User’s Engagement Influence Purchase Intention?. Electronics12(1), 122.

The authors are welcome to include these citations or from other researchers to provide better support for the justification.

·       The decision to exclude certain participants' data due to artifacts or incomplete sessions is understandable but also problematic. It's unclear whether this exclusion might introduce selection bias, favoring certain neural profiles over others. This uncertainty might challenge the representativeness of the sample and, by extension, the applicability of the findings to a broader population.

·       The paper doesn't discuss in depth how the data was cleaned before analysis, aside from mentioning the removal of artifacts. The handling of outliers, the process for dealing with missing or noisy data, and the criteria for data segmentation could significantly influence the results and should be transparently addressed.

·       Lines 435-436: The citations for the literature values of the mu band frequency during MI task are missing. Lines 351-352: The citations from recent research are missing, making the information presented not comprehensive and slightly outdated. I suggest adding more citations from recent research at both places. Please refer the following example:

·       Lakshminarayanan, K., Shah, R., Daulat, S. R., Moodley, V., Yao, Y., & Madathil, D. (2023). The effect of combining action observation in virtual reality with kinesthetic motor imagery on cortical activity. Frontiers in Neuroscience, 17, 1201865.

The authors are welcome to include these citations or from other researchers to provide better support for the justification.

·       The limitations section acknowledges key constraints but doesn't explore their implications thoroughly. For instance, what are the consequences of not being able to determine the direction of information flow? How might the results be interpreted differently if this limitation were addressed? Delving into these questions would demonstrate a critical understanding of the research.

·       The conclusion could also discuss future research directions more explicitly. What are the next logical steps in this line of investigation? Are there specific challenges that future studies need to address? Providing this guidance could help shape subsequent research efforts and advance the field.

In summary, while the paper presents innovative research with the potential to impact BCI technology significantly, there are several areas where the authors could expand their discussion, provide more detail, and engage more critically with their methods and findings. Doing so would strengthen the paper's credibility, relevance, and accessibility to a broader audience.

Reviewer 2 Report

Comments and Suggestions for Authors

The authors have proposed a methodology for classifying mental states based on the recording of EEG signals and amplitude modulation features. On one hand, the contribution suggested by the authors as relevant features for classification is not entirely original (amplitude modulation features), as other authors have used them in a similar manner. In the manuscript, this methodology is not well described and needs to be formulated/described more appropriately. This will help determine the methodological originality, or lack thereof, that the authors implicitly claim. On the other hand, in terms of the functional aspects of BCI technology, I doubt that the classification techniques employed can be implemented in real-time. Ultimately, BCI research aims to increase the speed of communication in these systems. However, the approach proposed can also be seen as a methodology for studying the underlying phenomena in mental tasks. In this context, I believe the contribution would be of great value. Nevertheless, the proposed title does not reflect this objective. Therefore, I suggest that the authors clarify the goals of their research and potentially change the title of their work.

Very important: this paper requires a general reorganization of its content.

Fig. 1 is not necessary.

Was the data from channels P8 and O1 removed for all subjects? Is the signal bad in all of them?

The FastICA algorithm was applied without taking into account channels P8 and O1. Subsequently, artifacts were removed using the ADJUST algorithm. Regarding this, some parameters should have been set for automatic removal. In the interest of result reproducibility, the authors should provide a more detailed description of how this procedure was carried out. For the purpose of my review, I would like to understand this procedure to determine whether this preprocessing stage was, or was not, implemented correctly. If possible, I wish to assess the implemented codes. The removal of artifacts in proposals/investigations of this kind is crucial and has a significant impact on the final results.

In this work, the authors propose the Amplitude Modulation Power Features; however, similar techniques have already been proposed by other authors, such as, for example, https://doi.org/10.3389/fncom.2017.00115. I suggest that the authors describe and formulate in more detail the procedure they have implemented for the extraction of this feature. Using mathematical formulations makes it possible to better visualize the originality of the method while also replicating the results obtained here.

In determining connectivity characteristics, the authors state that only 10 channels were used (to reduce computational demand and excessive dimensionality). Wouldn't it have been advisable to perform a pre-analysis to ensure that connectivity with the removed channels does not contribute to classification information? Then after such a pre-analysis, remove the channels that indeed do not provide information.

I believe that Fig. 3 and what is described in section 2.4 (Feature Selection, Classification, and Figures-of-Merit) are part of the results section. I suggest reorganizing the materials and methods sections and the results section.

Line 190, What are LFD scores?

On line 225 of the Results section, emerging results from methods that were not described in the Materials and Methods section are presented. I suggest that you accurately outline the methods used in their respective section.

In light of the aforementioned, I cannot recommend this work in its current state. I encourage the authors to enhance the organization of their research and clarify the objectives they are aiming to achieve. 

Comments on the Quality of English Language

-

Reviewer 3 Report

Comments and Suggestions for Authors

This article investigates the classification of seven mental tasks based on EEG amplitude modulation dynamic features. The characteristics of EEG features under various mental tasks are analyzed experimentally, and the classification efficiency of the method proposed in the article is demonstrated. The research work has implications for the study of brain-computer interfaces. However, there are still some minor issues that need to be explained:

1. Is the order of the bands under "Modulated Band" in the middle of Fig. 2 correct? Shouldn't the order of the bands be the same as that in the upper part of the figure, which should be delta, theta, alpha, beta, gamma?

2. 2.4 "As can be seen, performance tends to plateau after 2000 features for all feature type combinations." How is this 2000 determined? Are there any metrics?

Round 2

Reviewer 1 Report

Comments and Suggestions for Authors

Thank you for diligently addressing all the comments and making appropriate changes to the manuscript.

Reviewer 2 Report

Comments and Suggestions for Authors

The authors have appropriately addressed the reviews conducted. It is evident to me that the protocols and recorded data have already been evaluated and analyzed by them in previous works. I highly appreciate the provision of the pre-processing code (GitHub repository). Although some minor issues have arisen for me, I see no obstacle to accepting this work for publication.